# Short or Irregular Sleep Duration in Early Childhood Increases Risk of Injury for Primary School-Age Children: A Nationwide Longitudinal Birth Cohort in Japan

**DOI:** 10.3390/ijerph18189512

**Published:** 2021-09-09

**Authors:** Takafumi Obara, Hiromichi Naito, Kohei Tsukahara, Naomi Matsumoto, Hirotsugu Yamamoto, Takashi Yorifuji, Atsunori Nakao

**Affiliations:** 1Department of Emergency, Critical Care, and Disaster Medicine, Okayama University Graduate School of Medicine, Dentistry, and Pharmaceutical Sciences, 2-5-1, Shikata, Okayama 700-8558, Japan; tkfmobr16@okayama-u.ac.jp (T.O.); hei.trp@gmail.com (K.T.); hiro117_30@hotmail.com (H.Y.); qq-nakao@okayama-u.ac.jp (A.N.); 2Department of Epidemiology, Okayama University Graduate School of Medicine, Dentistry, and Pharmaceutical Sciences, 2-5-1, Shikata, Okayama 700-8558, Japan; naomi.f@nifty.com (N.M.); yorichan@md.okayama-u.ac.jp (T.Y.)

**Keywords:** sleep habits, trauma, problematic behavior, longitudinal study

## Abstract

The aim of this study was to investigate the longitudinal relationship between shorter or irregular sleep duration (SD) in early childhood and increased risk of injury at primary school age using data from a nationwide survey in Japan. We categorized SD into seven groups: 6 h, 7 h, 8 h, 9 hrs, 10 or 11 h, >12 h, and irregular, based on questionnaire responses collected at 5.5 years old. The relationship between SD and incidence of injury at 5.5–nine years of age is shown. In addition, we completed a stratified analysis on children with or without problematic behavior at eight years old. We included 32,044 children, of which 6369 were classified as having an injury and 25,675 as not having an injury. Logistic regression model showed that shorter or irregular SD categories were associated with an increased adjusted odds ratio (aOR) for injuries (6 h: aOR 1.40, 95% confidence interval (CI) 1.19–1.66, 7 h: aOR 1.10, 95% CI, 0.98–1.23, 8 h: aOR 1.13, 95% CI, 1.02–1.26, irregular: aOR 1.26, 95% CI 1.10–1.43). The same tendency was observed with shorter or irregular SD in subgroups with or without behavioral problems. Shorter or irregular sleep habits during early childhood are associated with injury during primary school age.

## 1. Introduction

Short or irregular sleep causes decreased activity of the prefrontal cortex and consequently inhibits concentration and behavioral and emotional control, leading to hyperactivity, depression, and poor communication [1,2]. In addition, insufficient sleep or bedtime delay in adolescents and children is associated with a higher risk of poor academic performance due to early memory and cognitive decline [3,4,5]. Multiple cohort studies also demonstrate that shorter sleep duration (SD) is associated with a higher risk of overweight/obesity and hypertension, as well as migraine and tension-type headaches in children and adolescents [6,7,8,9].

Previous studies have shown that insufficient/irregular sleep influences many factors, including mental and physical condition, life, and behavioral problems in children and adolescents [10,11,12]. Children with shorter SD are associated with more medically attended injuries [13] and accidental falls [14]. Meanwhile, Owens et al. found no relationship between SD and injury risk [15]. The relationship between short or irregular SD in childhood and risk of injury has not been fully elucidated due to various confounding factors involved in studies on injury in children and the limited sample sizes of single-center, cross-sectional studies [10].

In this study, we investigated the longitudinal relationship between shorter or irregular SD at 5.5 years of age and increased risk of injury in children during the primary school age years (5.5–9 years old) using data from a longitudinal, large, population-based, nationwide survey in Japan.

## 2. Materials and Methods

### 2.1. Participants

The Okayama University Ethics Committee approved this study (K1506-073) and waived the requirement for written informed consent. Study results are presented according to the STROBE guidelines for observational studies. This longitudinal study used data from “Longitudinal Survey of Babies in the Twenty-first Century,” a large birth cohort study conducted by the Japanese Ministry of Health, Labor, and Welfare (MHLW). The MHLW began conducting this survey annually in 2001, collecting data on children’s health, developmental status, and family circumstances from families all over Japan [16,17,18]. The survey targeted all babies born in Japan from January 10–17 or July 10–17 in 2001. When these babies reached six months old, the first surveys were randomly mailed to 53,573 families. Of these, guardians from 47,015 families originally competed and returned the questionnaires, yielding an 88% response rate. Participating families received follow-up questionnaires once every year. Birth records were also linked to each child’s survey data.

### 2.2. Sleep Duration (SD) and Irregular Sleep

SD was determined from responses to the optional questions “What time do you usually get up?” and “What time do you usually go to bed?” in follow-up questionnaires given at 5.5 years old. SD was divided into seven groups: 6 h, 7 h, 8 h, 9 h, 10 or 11 h, >12 h, and irregular. Irregular sleep was defined as no regular bedtime and wakeup schedule based on the responses from the returned questionnaires. The children with irregular sleep got different amounts of sleep each night.

The National Sleep Foundation and the American Academy of Sleep Medicine recommend that preschool-age children who are three to five years old get about 10–13 h of sleep daily. At this age, naps may get shorter, or a preschooler may stop routine napping [19]. However, some studies have shown significant differences in bedtimes and total SD among children from diverse regions [20]. In this study, “10 or 11 h” of SD at 5.5 years of age was defined as the standard SD in accordance with the MHLW sleep guidelines.

### 2.3. Injury (Outcomes)

“Injury” was defined based on answers to questions regarding injuries incurred from 5.5 to nine years old: “During the past year, did you have any injury for which you received treatment at the Hospitals, Clinics, or Other types of medical facilities?” The “Injury” group was defined as having an injury at least once during the 3.5 years study period, which was confirmed three times by the questionnaires. The “No Injury” group was defined according to responses indicating no injury on the questionnaire.

### 2.4. Statistical Analysis

We first categorized SD into these seven groups: 6 h, 7 h, 8 h, 9 h, 10 or 11 h, >12 h, and irregular. To study the effect of loss to follow-up, we compared baseline characteristics among eligible children included in the analysis (from the seventh to ninth surveys) and children lost to follow-up. After excluding those who lacked injury data, we investigated the seven groups of different SDs and their association with the clinical characteristics and outcomes using a descriptive analysis. We also compared baseline characteristics between participants with and without injuries.

Continuous variables and ordinal variables are described using medians with interquartile ranges; categorical variables are described using numbers and percentages. In this study, we adjusted for the following socioeconomic and biological factors: gender; term or preterm birth (<37 weeks of gestation); singleton or multiple; parity, including delivery of the child (one, two, ≥three); daycare attendance at 18 months old; maternal age at delivery (<25, 25–29, 30–34, ≥35 years old); maternal smoking; maternal educational attainment (university or higher (≥four years), junior college (two years), high school, or other); family income at seven years old (<four, four–seven, seven–10, or >10 million yen); and residential area (wards (district), cities, towns, or villages). Primary outcome was having an injury at least once during the study period. We used logistic regression analysis to examine the association between SD at 5.5 years old and the outcome. In the logistic regression analysis, 10–11 h SD was set as the reference group. Results of the logistic regression analyses were reported using the crude and adjusted odds ratio (OR) and 95% confidence interval (CI) for each SD. 

In addition, as a secondary outcome, we conducted a stratified analysis on children with or without problematic behavior at the age of eight years old [21]. Problematic behavior was defined based on whether answers to any of the following seven questions in the eighth survey was “yes”: (a) attention problems: (1) interrupting people, (2) inability to wait for their turn during play, and (3) failure to pay attention to surroundings when crossing the street; or (b) delinquent/aggressive behaviors: (1) lying, (2) destroying toys and/or books, (3) hurting other people, and (4) causing disturbances in public. Logistic regression analysis was also used to examine adjusted OR (aOR) and 95% CI. The same variables were used for adjustment.

All statistical analyses were performed with Stata version 15 statistical software (Stata-Corp LP, College Station, TX, USA).

## 3. Results

### 3.1. Participant Characteristics

Figure 1 shows the study’s flow diagram. We targeted data on children collected from the sixth survey (given at 5.5 years old) to the ninth survey (given at nine years old). We calculated SD from questionnaire data obtained at 5.5 years old and enrolled 37,709 children in the study. Then, we extracted information about injuries from the seventh to ninth surveys (children seven to nine years old). Children missing information about the injury were excluded. By the ninth survey, 5665 children were lost to follow-up, leaving 32,044 children that were included in the analysis.

The flow diagram (Figure 1) reports data included in the analysis about injury in 32,044 participants; 6369 (19.9%: 6369/32,044) children were categorized as having had an injury and 25,675 (80.1%: 25,675/32,044) were categorized as having had no injury.

Table 1 shows the baseline characteristics of the participants, including children who were included in the analysis or lost to follow-up.

Children missing information about their injury in the seventh to ninth surveys were more likely to have short or irregular SD. Their mothers were relatively younger and were more likely to be smokers and less educated, and they were more likely to be missing data regarding income and problem behavior compared with children included in this analysis.

Then, we summarized the characteristics of children with data available about the injury (Table 2).

We found no significant difference in characteristics between the injury group and the no injury group, except that more boys were in the injury group.

### 3.2. Sleep Duration (SD) and Injury 

The relationships between all SD categories and injuries in follow-up questionnaires at seven to nine years old using univariable and multivariable logistic regression analysis are shown in Table 3. 

Only 2921 (9.1%: 2921/32,044) children had the standard SD as recommended by the MHLW (10 or 11 h). Of all the SD categories, the 8 h group was both the largest and had the highest percentage of injuries (injury-case 35.4%: 2260/6369). We observed a strong gradient in risk of visits to medical institutions according to shorter and irregular SD in the crude model. Additionally, the same category of SD was associated with increased OR for injuries in the adjusted model (6 h: aOR 1.40, 95% CI, 1.19–1.66, 7 h: aOR 1.10, 95% CI, 0.98–1.23, 8 h: aOR 1.13, 95% CI, 1.02–1.26, irregular: aOR 1.26, 95% CI 1.10–1.43). Furthermore, it was suggested that ORs tended to decrease as the amount of sleep approached 10 or 11 h.

### 3.3. Sleep Duration and Problematic Behavior (Stratified Analysis)

Association between SD and injury with or without problematic behavior at eight years of age is shown in Table 4. 

In the group with no problematic behavior (n = 12,214), shorter or irregular SD was associated with injury during the period (6 hrs: aOR 1.39, 95% CI, 1.04–1.84, 7 h: aOR 1.25, 95% CI, 1.03–1.52, 8 h: aOR 1.20, 95% CI, 1.00–1.44, irregular: aOR 1.49, 95% CI, 1.18–1.87). In the problem behavior group (n = 18,408), the same tendency was observed in the 6 hrs and irregular SD groups (6 h: aOR 1.38, 95% CI, 1.12–1.71, irregular: aOR 1.20, 95% CI, 1.01–1.42).

## 4. Discussion

The present study demonstrates that shorter or irregular SD in early childhood was closely associated with a higher risk of injury at school age using data from a nationwide Japanese survey. This association did not change, even after we adjusted for an extensive list of potential confounders in the stratified analysis. To our knowledge, this is the largest study showing a clear longitudinal relationship between shorter or irregular SD in early childhood and increased risk of injury during the primary school age years. 

Previous longitudinal studies have highlighted the relationship between sleep quality in early childhood and adverse events such as overweight/obesity, hypertension, and behavioral problems at school age [7,8,9,22,23]. A strength of the present study is that it used a large-scale, population-based, nationwide database. The survey collected data from more than 50,000 births, 1/20th of the number of births in Japan in 2001. Our number of study participants is larger than that of previous studies; thus, we learned about more accurate trends among participants in Japan. In addition, our very high baseline response rate strengthens the validity of our findings. The large sample size and response rate more clearly shows the relationship between SD in early childhood and injury at early school age. 

Some studies suggest that daytime functioning may be disturbed by even minor sleep restrictions in children [24]. Meanwhile, a UK cohort study indicated that the effects of not having regular bedtimes on adverse behavioral outcomes were reversible [25]. Based on our study, we should emphasize the importance of appropriate sleep during young childhood to parents and caregivers. Our results indicate that the whole living environment, including appropriate sleep habits in early childhood, may have significant implications for child health and public health.

We categorized SD at five years and six months, even though children change their habits and sleep relatively less as they begin their school life. In addition, they have a wider range of activities and are more likely to work alone, which increases the risk of injuries. We had concerns regarding whether the amount of SD in early childhood really affects the risk of injury at school age. However, in fact, cerebrums have generally developed, sleep-awake patterns have been established, and naps may get shorter or stop in children older than six years of age [19]. Additionally, Kelly et al. examined the relationship between bedtime habits and school-age behavior at ages three, five, and seven years old. Their results showed that irregular sleep habits in early childhood show an effect by school age [25]. Comparison of findings from any nation-specific studies may be difficult, as all have used varying age groups and included different survey questions with varying sleep definitions. Our results were consistent with those from previous reports showing the importance of regular bedtimes in infancy.

Some studies have reported that disrupted sleep is associated with a range of problematic behavior during school age (attention-deficit/hyperactivity disorder and aggressive behavior) [22,26]. Using the same database from the MHLW, Kobayashi et al. found that poor toddler-age sleep duration predicted behavioral problems during early school age (at eight years old) [21]. Our stratified analysis showed that shorter or irregular sleep schedules in early childhood were closely associated with an increased risk of injury at school age after excluding the effects of problematic behavior at eight years old (Table 4). It has been suggested that various biological rhythm factors are disturbed by irregular lifestyles and late waking and bed times [27]. Therefore, we concluded that not only the problem behavior, but also short or irregular SD may itself cause inattention and poor concentration, which may increase the risk of seeking medical attention due to injury.

In general, it is important to establish confounding factors and conduct multivariate analyses based on the premise of organizing relevant factors in multivariate analysis and understanding their characteristics. The results of characteristics about children lost to follow-up shown in Table 1 were easy for us to accept; those children were more likely to have short or irregular SD, had more missing data regarding income and problem behavior, and their mothers were relatively younger, smokers, and less educated compared with children included in the study. Insufficient sleep might indicate a difficult family environment, but we could not confirm this. In addition, the large number of children with short or irregular SD in the excluded groups might underestimate the actual results.

### Limitations

First, SD was calculated from the difference between the times of falling asleep and waking and did not consider mid-wake or nap times. Moreover, the questionnaire sleep schedules suggested “usual” schedules; therefore, we speculated that most families reported data only on their children’s weekday sleep schedules. Subsequently, responses may not reflect the exact duration of sleep in a day. Secondly, since there was only a “yes” or “no” question regarding injuries, the location, type, extent, and severity of injuries were not known. Third, SD was obtained only once at the age of 5.5. However, sleep habits are established at this age; SD afterwards could be inferred. Problematic behavior was reported only at the age of eight years old. Additionally, information regarding SD was absent at the ages of seven, eight, or nine years old in the MHLW database. Finally, the content of the MHLW database depended largely on the subjectivity and memory of the parents answering the questionnaire each year.

Despite these limitations, the results of this study using a national survey are highly significant because of the large sample size and the results after adjusting for a wide variety of questionnaire items. In addition, it is difficult to adjust for the effects of various confounding factors related to children’s injuries by identifying environmental factors from interviews conducted in the time-limited emergency department. The only way to show a causal relationship with the environmental factors of growth and development over time is to conduct a longitudinal cohort study with long-term follow-up of the same and multiple individuals. Our study overcomes these problems, and the results add objective new findings to inspire future sleep research. 

## 5. Conclusions

In the present study, we showed that insufficient or irregular sleep duration in early childhood increases the risk of medical consultation due to injury during school age using a Japanese national database. In addition, the same tendency was observed in children with and without problematic behavior.

## Figures and Tables

**Figure 1 ijerph-18-09512-f001:**
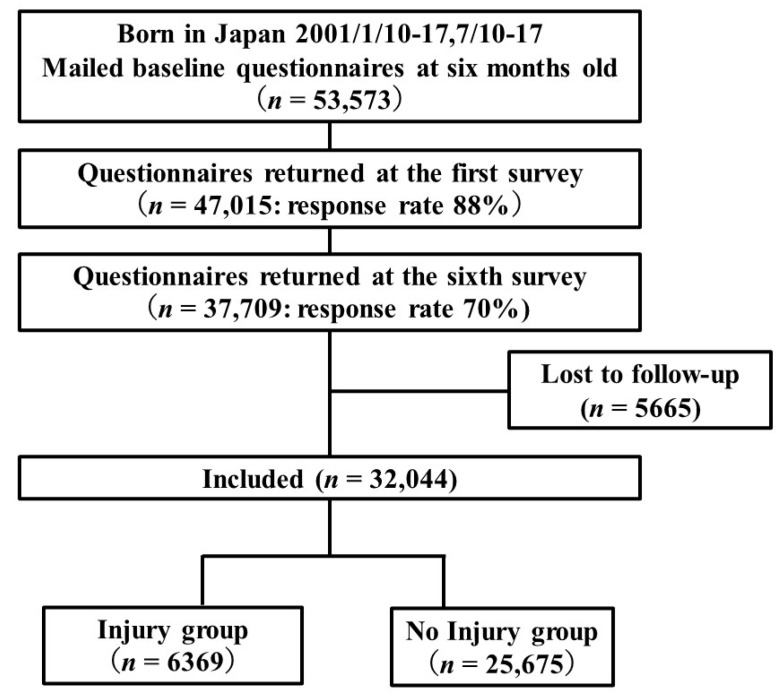
Flow diagram for the selection of participant children from birth to seven to nine years old.

**Table 1 ijerph-18-09512-t001:** Characteristics of children included in the study and lost to follow-up at 5.5–9 years old (*n* = 37,709).

	Included (*n* = 32,044)	Lost to Follow-Up (*n* = 5665)
Sleep Duration, *n* (%)		
6 h	1191 (3.7)	269 (4.7)
7 h	6773 (21.1)	1224 (21.6)
8 h	11,249 (35.1)	1926 (34.0)
9 h	7280 (22.7)	1182 (20.9)
10 or 11 h	2921 (9.1)	423 (7.5)
>12 h	52 (0.2)	10 (0.2)
Irregular	2578 (8.0)	631 (11.1)
Gender, *n* (%)		
Male	16,682 (52.1)	2911 (51.4)
Female	15,362 (47.9)	2754 (48.6)
Term or preterm birth, *n* (%)		
Term	30,515 (95.2)	5360 (94.6)
Preterm	1529 (4.8)	305 (5.4)
Singleton or multiple birth, *n* (%)		
Singleton	31,418 (98.0)	5532 (97.7)
Multiple	626 (2.0)	133 (2.3)
Parity, *n* (%)		
1 (no older siblings)	15,650 (48.8)	2707 (47.8)
2	11,883 (37.1)	2033 (35.9)
3	4511 (14.1)	925 (16.3)
Daycare Attendance, *n* (%) *		
Nursery	26,733 (83.4)	4450 (78.6)
Home	4944 (15.4)	961 (17.0)
Missing Data	367 (1.1)	254(4.5)
Maternal Age at Delivery, *n* (%)		
<25 (years old)	3029 (9.5)	934 (16.5)
25–30	12,153 (37.9)	2192 (38.7)
30–35	12,240 (38.2)	1853 (32.7)
>35	4622 (14.4)	686 (12.1)
Maternal Smoking, *n* (%)		
None	27,753 (86.6)	4354 (76.9)
Yes	4163 (13.0)	1272 (22.5)
Missing Data	128 (0.4)	39 (0.7)
Maternal Education, *n* (%)		
University or Higher	4873 (15.2)	551 (9.7)
Junior College	13,711 (42.8)	2051 (36.2)
High School or Other	12,975 (40.5)	2800 (49.4)
Missing Data	485 (1.5)	263 (4.6)
Family income at Seven Years Old, *n* (%)		
<400 (million Yen)	4042 (12.6)	489 (8.6)
400–700	13,533 (42.2)	1145 (20.2)
700–1000	6768 (21.1)	434 (7.7)
>1000	3626 (11.3)	242 (4.3)
Missing Data	4075 (12.7)	3355 (59.2)
Residential Area, *n* (%)		
Wards (Districts)	7088 (22.1)	1187 (21.0)
Cities	18,876 (58.9)	3355 (59.2)
Towns or Villages	6080 (19.0)	1123 (19.8)
Problematic Behavior, *n* (%) **		
No	12,214 (38.1)	888 (15.7)
Yes	18,408 (57.4)	1556 (27.5)
Missing Data	1422 (4.4)	3221 (56.9)

* Obtained from the second survey (at 18 months of age). ** Obtained from the eighth survey (at eight years of age).

**Table 2 ijerph-18-09512-t002:** Characteristics of children included in the analysis, with and without injuries (*n* = 32,044).

	Injury (*n* = 6369)	No Injury (*n* = 25,675)
Gender, *n* (%)		
Male	3907 (61.3)	12,775 (49.8)
Female	2462 (38.7)	12,900 (50.2)
Term or Preterm Birth, *n* (%)		
Term	6036 (94.8)	24,479 (95.3)
Preterm	333 (5.2)	1196 (4.7)
Singleton or Multiple Birth, *n* (%)		
Singleton	6271 (98.5)	25,147 (97.9)
Multiple	98 (1.5)	528 (2.1)
Parity, *n* (%)		
1 (no older siblings)	3035 (47.7)	12,615 (49.1)
2	2466 (38.7)	9417 (36.7)
3	868 (13.6)	3643 (14.2)
Daycare Attendance, *n* (%) *		
Nursery	5292 (83.1)	21,441 (83.5)
Home	1014 (15.9)	3930 (15.3)
Missing Data	63 (1.0)	304 (1.2)
Maternal Age at Delivery, *n* (%)		
<25 (years old)	536 (8.4)	2493 (9.7)
25–30	2453 (38.5)	9700 (37.8)
30–35	2473 (38.8)	9767 (37.8)
>35	907 (14.2)	3715 (14.5)
Maternal Smoking, *n* (%)		
None	5553 (87.2)	22,200 (86.5)
Yes	794 (12.5)	3369 (13.1)
Missing Data	22 (0.3)	106 (0.4)
Maternal Education, *n* (%)		
University or Higher	1014 (15.9)	3859 (15.0)
Junior College	2871 (45.1)	10,840 (42.2)
High School or Other	2392 (37.6)	10,583 (41.2)
Missing Data	92 (1.4)	393 (1.5)
Family income at Seven Years Old, *n* (%)		
<400 (million Yen)	750 (11.8)	3292 (12.8)
400–700	2731 (42.9)	10,802 (42.1)
700–1000	1418 (22.3)	5350 (20.8)
>1000	714 (11.2)	2912 (11.3)
Missing Data	756 (12.0)	3319 (12.9)
Residential Area, *n* (%)		
Wards (Districts)	1490 (23.4)	5598 (21.8)
Cities	3745 (58.8)	15,131 (58.9)
Towns or Villages	1134 (17.8)	4946 (19.3)

* Obtained from the second survey (at 18 months of age).

**Table 3 ijerph-18-09512-t003:** Logistic regression analysis to investigate the relationship between sleep duration and injuries at 5.5–9 years old.

	Case/*n* (%)	Crude OR (95% CI)	Adjusted OR * (95% CI)
Sleep Duration			
6 h	282/1191 (23.7)	1.39 (1.18–1.64)	1.40 (1.19–1.66)
7 h	1334/6773 (19.7)	1.10 (0.98–1.23)	1.10 (0.98–1.23)
8 h	2260/11,249 (20.1)	1.13 (1.01–1.25)	1.13 (1.02–1.26)
9 h	1388/7280 (19.1)	1.06 (0.94–1.18)	1.06 (0.95–1.19)
10 or 11 h	533/2921 (18.2)	1 (Reference)	1 (Reference)
>12 h	11/52 (21.2)	1.20 (0.61–2.35)	1.23 (0.62–2.41)
Irregular	561/2578 (21.8)	1.25 (1.09–1.42)	1.26 (1.10–1.43)

*n*, number of children; OR, odds ratio; CI, confidence interval; * Adjusted for gender, term or preterm birth, singleton or multiple birth, parity, daycare attendance, maternal age at delivery, maternal smoking, maternal education, income, and residential area.

**Table 4 ijerph-18-09512-t004:** Logistic regression analysis to investigate the relationship between sleep duration and injury stratified by presence or absence of problematic behavior at eight years old.

	No Problematic Behavior (*n* = 12,214)		Problematic Behavior (*n* = 18,408)	
	Case/*n* (%)	Adjusted OR * (95% CI)	Case/*n* (%)	Adjusted OR * (95% CI)
Sleep Duration				
6 h	98/464 (20.0)	1.39 (1.04–1.84)	170/673 (25.3)	1.38 (1.12–1.71)
7 h	497/2640 (18.8)	1.25 (1.03–1.52)	778/3829 (20.3)	1.04 (0.90–1.21)
8 h	795/4383 (18.1)	1.20 (1.00–1.44)	1388/6417 (21.6)	1.14 (1.00–1.30)
9 h	443/2711 (16.3)	1.06 (0.87–1.29)	864/4226 (20.4)	1.06 (0.92–1.22)
10 or 11 h	171/1.087 (15.7)	1 (Reference)	330/1683 (19.6)	1 (Reference)
>12 h	5/23 (21.7)	1.63 (0.59–4.48)	6/26 (23.1)	1.18 (0.47–2.99)
Irregular	194/906 (21.4)	1.49 (1.18–1.87)	349/1554 (22.5)	1.20 (1.01–1.42)

*n*, number of children; OR, odds ratio; CI, confidence interval; * Adjusted for gender, term or preterm birth, singleton or multiple birth, parity, daycare attendance, maternal age at delivery, maternal smoking, maternal education, income, and residential area.

## Data Availability

The data for the study were obtained from a large birth cohort study conducted by the MHLW: Longitudinal Survey of Babies in the Twenty-first Century; the authors do not have permission to share data.

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
