# Peer review of "Short or Irregular Sleep Duration in Early Childhood Increases Risk of Injury for Primary School-Age Children: A Nationwide Longitudinal Birth Cohort in Japan"

_ijerph, 2021, doi:10.3390/ijerph18189512_

Round 1
Reviewer 1 Report
Brief Summary:
This is a longitudinal study that examines the relationship between short or irregular sleep duration in early childhood to assess if there is a relationship with risk of injury for primary school-aged children using a nationwide longitudinal birth cohort in Japan. It showed that insufficient or irregular sleep duration in early childhood increased the risk of injury in primary school-aged children.
Broad Comments:
As correctly stated by the authors the strengths of this study include use of a longitudinal approach, and use of a large-scale population based nationwide database. The study is interesting but has significant short comings.
Specific Comments:
- Irregular Sleep first appears in the title of the manuscript. The definition of “irregular sleep” is not provided in the Introduction or Materials and Methods sections or, for that matter, anywhere in the manuscript. There is no information about how it is scored either. If there is no definition or information about how to score it them it probably should be removed from the manuscript altogether.
- Introduction: Irregular sleep as it relates to injuries has not been addressed in the introduction. A secondary outcome – behavior problems, is mentioned at line 107; however, behavior problems have not been discussed in the Introduction section. For example how are behavior problems related to sleep duration and injuries? The description of behavior problems (line 108-113) is suggestive of ADHD, conduct disorders and other externalizing conditions. Is this what the authors have in mind? If so then it has also not been addressed in the Introduction section.
- Material and Methods:
- If I have read it correctly then sleep duration was estimated only once at age 5 and a half years and not over the subsequent years 6, 7, 8 or 9 years of age. Is that correct?
- Again if I have read it correctly then injuries were estimated yearly for the 3.5 years duration of the study. Is that correct?
- Is there any reason sleep duration was not estimated yearly as were injuries? A fairly similar study published in 2017 did just that- Marlenga, B., King, N., Pickett, W., Lawson, J., Hagel, L., Dosman, J. A., & Saskatchewan Farm Injury Cohort Study Team. (2017). Impact of sleep on injury risk among rural children. Paediatrics & child health, 22(4), 211-216. I believe is should be possible for the authors to estimate sleep duration during the 3.5 year study period and one wonders why this was not done.
- It is not clear the basis for selecting problematic behavior only at the age of 8 years. Could the authors explain why this was not done for the other years?
- Discussion:
- We do not know if sleep duration was present or absent at age, 7, 8 or 9 years. That being the case it is difficult to make an association between sleep duration and injuries. Therefore one could argue that the association as indicated by the authors is not justifiable, as opposed to the Marlenga et al (2017) article. Perhaps one could try to make an association for the first year only involving the 5 and a half year olds and injuries in the previous year for that group only.
- The same deficiency applies to trying to the attempt to link sleep duration, problem behaviors and injuries at age 8 years when there is no information about sleep duration during the age of 8 years.
Other comments:
- Line 49: This line is unclear. I suspect …risk of injury (during) in children during….. was meant to read…risk of injury in children during the primary…The authors should correct this if it is a typographical error.
- Line 67 to 70. The way these lines are written would suggest it was the young children who completed the measures when in fact it was parents/families which did. They should be re-written to bring them in line with lines 236-237.
Reviewer 2 Report
Dear Author,
I am very happy to read Your well written manuscript, however I found some concerns about it:
1 It could be useful if You revise Your references section in order to add latest references
2 Why You evaluate for Your analyses only SD at 5.5 age of years and not also the SD parameters at all surveys? This point is not clearly described in the text
Regards
